# V1- and IT-like representations are directly accessible to human visual perception

**Akshay V. Jagadeesh**
Department of Psychology
Wu Tsai Neurosciences Institute
Stanford University
akshayj@stanford.edu

**Justin L. Gardner**
Department of Psychology
Wu Tsai Neurosciences Institute
Stanford University
jlg@stanford.edu

## Abstract

Human visual recognition of complex patterns is supported by hierarchical representations in the ventral stream of visual cortex. However, it remains undetermined whether representations in early visual cortical areas, e.g. primary visual cortex (V1), are directly accessible by perception or are merely intermediates used only for the generation of more complex representations in higher-level visual areas, e.g. inferior temporal cortex (IT). Here, we constructed deep convolutional neural network (dCNN) based simulations of V1 and IT by linearly weighting dCNN features to maximize predictivity of electrophysiological responses. We used these cortical simulations to synthesize stimuli which linearly interpolate through either a V1- or IT-like feature space. In a visual discrimination task, we found that human observers are highly sensitive to variation through both V1 and IT representational spaces. We found that behavior on this task cannot be explained by an observer model that makes use of solely V1 features or IT features, but instead is best explained by a weighted combination of V1 and IT features. Our results thus provide evidence for the insufficiency of IT-like representations and the utility of representations in both early and late regions of the ventral visual stream for human visual perception.

## 1   Introduction

The ventral stream of primate visual cortex contains representations that increase both in selectivity for complex visual features (e.g. texture, shape, category) and in invariance to low-level visual features (e.g. contrast, rotation, retinotopic location) from primary visual cortex (V1) to inferior temporal cortex (IT) [1, 2]. While invariance is useful for generating category representations that are robust to identity-preserving transformations, it is nonetheless the case that human observers are perceptually sensitive to low-level visual features if asked [3, 4, 5, 6]. What is the nature of the perceptual readout from the ventral visual stream that enables human observers to recognize both complex and low-level visual features?

In one view, representations in IT are sufficient to account for a wide variety of human visual recognition behaviors [7, 8]. This view is supported by evidence that visual features such as color [9, 10], shape [11, 12, 10], texture [10, 13], pose, and rotation [14] can be accurately decoded from IT, in addition to category information [15, 16]. A contrasting view suggests that a perceptual readout must be able to directly access sensory representations in both early visual cortex and IT cortex to support the full breadth of human visual recognition capabilities. To substantiate this view, it is necessary to functionally dissociate the contributions of V1 representations from those in IT. However, dissociating the functions of early and late ventral stream areas is made more difficult by the finding

3rd Workshop on Shared Visual Representations in Human and Machine Intelligence (SVRHM 2021) of the Neural Information Processing Systems (NeurIPS) conference, Virtual.

that category-orthogonal visual features, such as pose, rotation, and scale, traditionally thought to be encoded by early visual cortex, can actually be decoded more easily from later ventral stream areas [14].

Here, instead of hand-selecting features and labeling them as low- or high-level, we employed a model-based approach to directly compare the representational geometry of V1 and IT to that of human visual perception. To do so, we simulated populations of neurons in two visual cortical areas, V1 and IT, and used these simulations both to synthesize stimuli and to model human behavior. To assess the contribution of V1- and IT-like representations to perception, we synthesized a stimulus set which linearly traversed through a V1-like representational space and another which linearly traversed through a IT-like representational space. We reasoned that if human observers only had direct access to IT representations, then they should be unable to accurately discriminate stimuli which varied linearly through V1 representational space but were highly nonlinear in IT representational space. Instead, we found that human observers were highly sensitive to variation through both V1- and IT-like representational spaces. We compared the performance of observer models which read features out of a V1 model only, IT model only, or used a joint readout from V1 and IT models, and found that the joint readout model best explained human behavioral performance. These results provide evidence that a state-of-the-art model of IT is insufficient to support human recognition of visual features that span through a V1-like feature space, suggesting the direct accessibility of both V1 and IT representations to perception.

## 2 Modeling V1 and IT cortical responses

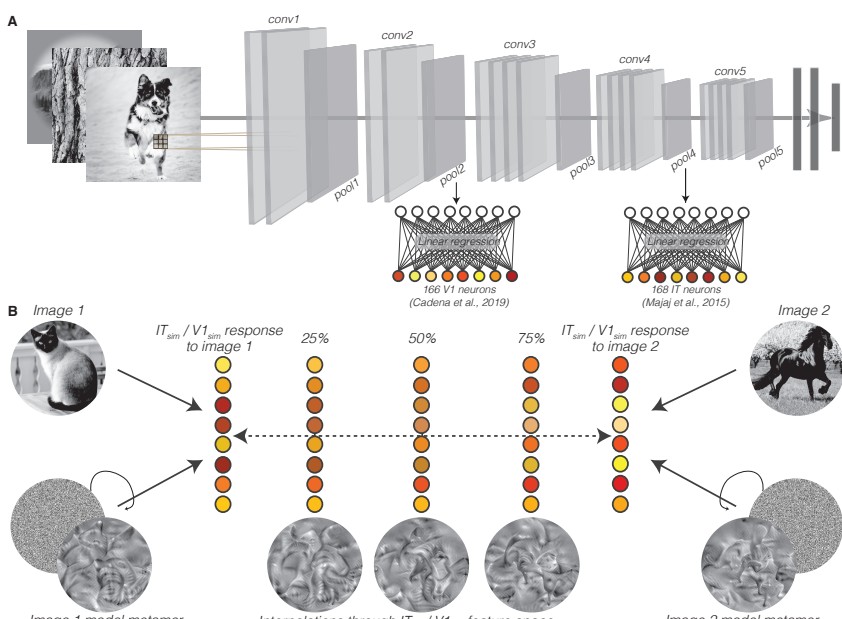

Figure 1: Using V1 and IT simulations to synthesize stimuli which linearly traverse through neural representations. (A) Deep convolutional neural network model of IT and V1. Imagenet-trained VGG-19 features from layer pool2 are fit to predict V1 neurons and features from layer pool4 are fit to predict IT neurons. (B) Metamer synthesis is performed by matching the $IT_{sim}/V1_{sim}$ response to a natural image. Image interpolation is performed by interpolating between the $V1_{sim}/IT_{sim}$ response to two different images and synthesizing images that match the intermediate interval responses.

We modeled the visual response properties of neurons in V1 and IT as a linear combination of features [17] from an Imagenet-trained VGG19 [18] dCNN model. Using publicly available published electrophysiology datasets of 166 units in V1 [19] and 168 units in IT ([8], [20]), we used partial least squares (PLS) linear regression to find a linear weighting of VGG19 features that maximized

the predictivity of each unit's firing rate in response to a large set of visual stimuli. We found that the layer which best predicted held-out neural responses in V1 was pool2 (mean $R^2 = 0.37$) and the layer which best predicted held-out neural responses in IT was pool4 (mean $R^2 = 0.51$). Thus, by multiplying the activations from those layers by weights, learned via linear regression, we were able to simulate the responses of a population of 166 V1 neurons and a population of 168 IT neurons (Fig. 1A). We will refer to these model neural population responses as $V1_{sim}$ and $IT_{sim}$, respectively.

# 3  Synthesizing stimuli to linearly traverse V1 and IT representational space

By matching either the $V1_{sim}$ or $IT_{sim}$ population response to a natural image, we were able to synthesize images which were model-metameric (i.e. pixel-wise different images which evoke identical simulated neural responses) in the $V1_{sim}$ or $IT_{sim}$ representational space, respectively (Fig. 1B). Although the term "metamer" is sometimes used to refer to two stimuli which are perceptually indistinguishable [21], in this case, we are using it to refer to two stimuli which evoke identical model responses but are quite perceptually distinct [22]. We used gradient descent to iteratively update the pixels of a randomly initialized image to minimize the mean squared error between the simulated neural response evoked by the synthesized image and the simulated neural response evoked by a natural image. We also incorporated a total variational loss to minimize high frequency noise in the synthesized outputs, an approach which has been used previously when synthesizing images by matching model responses [23, 24]. The resulting synthesized images were thus metameric to their source natural images in either the $V1_{sim}$ or $IT_{sim}$ representation. By initializing the synthesis procedure with a different random seed image, we generated multiple samples which all evoked nearly identical model responses, though they differed greatly in their pixel values.

To synthesize images which linearly traversed the $V1_{sim}$ or $IT_{sim}$ representation, we interpolated between the simulated neural response evoked by two different images, at 25%, 50%, and 75% intervals, and synthesized images to match the model neural response at each of those intervals between the model neural responses to two different natural images (Fig. 1B). That is, if $X_i$ is the vector of $IT_{sim}$ responses to image $i$ and $X_j$ is the vector of $IT_{sim}$ responses to image $j$, then we synthesized images to match the $IT_{sim}$ responses at $\alpha \times X_i + (1 - \alpha) \times X_j$ for $\alpha \in [0, .25, .5, .75, 1]$. We will refer to the set of images that linearly interpolate through $IT_{sim}$ as $IT - linear$ images and the set of images that linearly interpolate through $V1_{sim}$ as $V1 - linear$ images.

We selected 10 different images, including images of objects, textures, as well as sinusoidal gratings and for each image, we extracted the $V1_{sim}$ response and the $IT_{sim}$ response. To synthesize the $V1 - linear$ images, we interpolated between the $V1_{sim}$ response to two different images and synthesized 5 samples at each of 5 evenly spaced intervals (0%, 25%, 50%, and 75%, 100%, where 0% and 100% are model metamers of the two original images) between the two images. To synthesize $IT - linear$ images, we interpolated between the $IT_{sim}$ response to two different images at each of the same 5 evenly spaced intervals. See supplementary figure 1 for more example stimuli used in this experiment.

We validated that the resulting synthesized images indeed linearly interpolated through the feature space represented by the given population of model IT / V1 neurons. To do so, we passed each of the synthesized images back into the neural models and extracted the simulated neural response to each image. Then, we tested how well the trajectory of stimuli which were synthesized to interpolate from one image response to another could be explained by a linear regression over model neurons. We found that images generated to be linear in V1 (V1-linear images) were well-explained ($R^2 = 0.988$) by a linear function of $V1_{sim}$ neurons, and that images generated to be linear in $IT_{sim}$ (IT-linear images) were well-explained by a linear function of $IT_{sim}$ units ($R^2 = 0.997$). This was expected and validated that the synthesis process worked as intended. Additionally, we found that V1-linear images were highly nonlinear in the $IT_{sim}$ representational space ($R^2 = 0.368$) and that IT-linear images were highly nonlinear in the $V1_{sim}$ representational space ($R^2 = 0.080$). This latter finding is not necessarily expected, as it could have been that the V1-linear images were linear in the IT representational space, if for example the V1 representation was a subspace of the IT representational space.

# 4 How does human perceptual sensitivity relate to V1 and IT representations?

We performed a 2 alternative forced choice (2-AFC) task to measure perceptual sensitivity to stimuli which varied along dimensions of either the $IT_{sim}$ or $V1_{sim}$ feature space. On each trial, subjects were concurrently presented with a probe image and two choice images for up to 5 seconds (Fig. 2A). The probe image, presented centrally, had been synthesized to evoke a model neural response 0% (i.e. matched to image 1), 25%, 50%, 75% or 100% (i.e. matched to image 2) of the way between two different images for 1 second. The choice images, presented on the left and right sides of the display, were always 0% and 100% of the way between the two images (i.e. model metamer of a natural image), with the samples selected so that the choice images were never identical to the probe images. Subjects were instructed to select which of the two choice images appeared more similar to the probe image. On each trial, all three images varied linearly either in $V1_{sim}$ or in $IT_{sim}$ representational space. We examined the likelihood that human subjects and cortical observer models selected the 100% choice, as a function of the interpolation interval of the probe image.

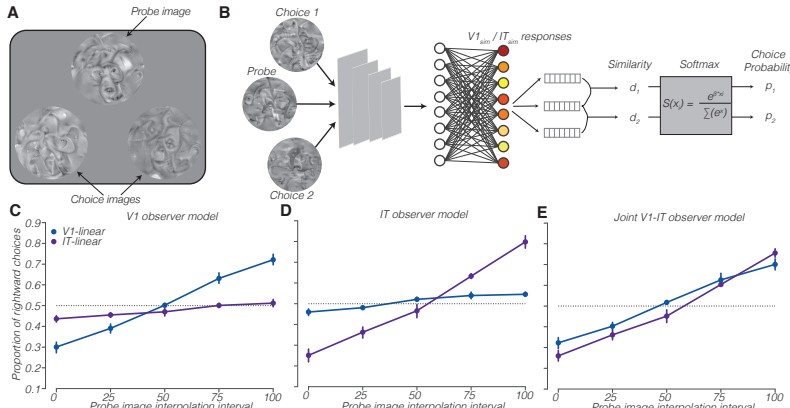

Figure 2: Comparison of $V1_{sim}$ and $IT_{sim}$ observer models yields predictions for human behavior. (A) 2-AFC task design. Probe image is presented at fixation, and choice images are presented to the left and right. Subject is asked to choose whether the probe image looks more similar to the left or right choice. (B) $V1_{sim}$ observer model is highly sensitive to V1-linear stimuli but mostly insensitive to IT-linear stimuli. (B) $IT_{sim}$ observer model is highly sensitive to IT-linear stimuli but weakly sensitive to V1-linear stimuli. (C) Joint readout model which weights $V1_{sim}$ and $IT_{sim}$ features is sensitive to both V1-linear and IT-linear stimuli.

## 4.1 Hypothesis: which cortical representations might support visual perception?

We evaluated the behavior of different observer models in this task to understand what performance could be expected if perception were supported by V1 alone, IT alone, or V1 and IT together. We constructed observer models which performed the 2-AFC similarity judgment task by computing the Pearson correlation between the simulated neural response to the probe image and that of each of the two choices, and transforming these representational similarities into choice probabilities using a softmax function, with a single exponent parameter that could be fit to the behavioral data (Fig. 2B). We compared the behavior of 3 different observer models which used different feature spaces to calculate the representational similarity - one with access to only $V1_{sim}$ features, one with access to only $IT_{sim}$ features, and one which weighted $V1_{sim}$ and $IT_{sim}$ features (this model had two additional free parameters: the weight assigned to the $V1_{sim}$ feature dissimilarity and the weight assigned to the $IT_{sim}$ feature dissimilarity). We found that the V1 observer model was highly sensitive to V1-linear stimuli and insensitive to IT-linear stimuli (Fig. 2C), while the IT observer model was highly sensitive to IT-linear stimuli but insensitive to the V1-linear stimuli (Fig. 2D). Only the joint V1-IT readout model was sensitive to both the V1-linear and IT-linear stimuli (Fig. 2B). Thus, if human observers are only sensitive to IT-linear images, we could reasonably conclude that human perception is only able to access IT features, whereas if human observers are sensitive to both

IT-linear and V1-linear images, we could conclude that perception must have direct access to the features represented in both $V1_{sim}$ and $IT_{sim}$.

## 4.2 Human perception is supported by a joint readout of both V1 and IT features

Human observers were highly sensitive to both V1-linear and IT-linear stimuli. We collected 3000 total trials of data across 18 subjects, recruited using Amazon Mechanical Turk. All protocols were approved beforehand by the Institutional Review Board for research on human subjects, and all observers gave informed consent prior to the start of the experiment. We assessed perceptual sensitivity by evaluating the likelihood of selecting the 100% choice (denoted as the "rightward" choice, even though 0% and 100% choices were left-right balanced) as a function of the interpolation interval of the probe image (Fig. 3A). We found that there was a significant effect of probe interpolation interval on the probability of selecting the 100% choice, for both V1-linear and IT-linear images ($\beta_{V1-interval} = 0.0783, 95\%CI = [0.055, 0.102], p < 0.001$, $\beta_{IT-interval} = 0.1182, 95\%CI = [0.095, 0.142], p < 0.001$).

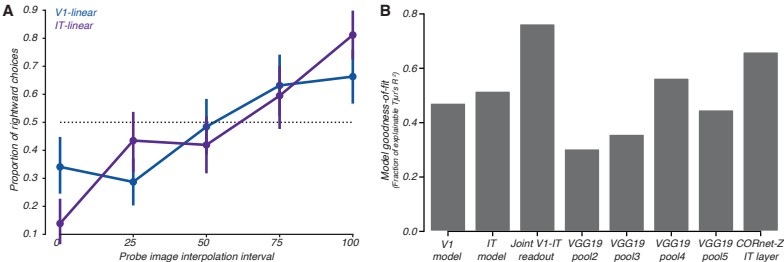

Figure 3: Task design and behavioral results. (A) Human behavioral performance as a function of the interpolation interval of the probe image, with V1-linear images in blue and IT-linear images in purple. (B) Comparison of models fit to human behavior reveals that joint V1-IT readout best predicts behavioral performance. Goodness-of-fit measured by Tjur's coefficient of discrimination [25].

We subsequently fit each of the three observer models described previously to the human behavioral data by estimating the free parameters to maximize the likelihood of the observed behavioral data, and evaluated the goodness-of-fit of the models on a held-out subset of the behavioral data, using cross-validated Tjur's $R^2$ [25] (Fig. 3B). Our results suggest that the model which used a weighted combination of both V1 and IT features fits human behavioral performance significantly better than both the V1 observer model (p=0.003) and the IT observer model (p=0.010). We also evaluated the performance of 5 additional observer models, which were constructed by using activations from different intermediate layers in VGG19 [18] or CORnet-Z [26] to compute the similarity between the probe image and each of the choice images on each trial. Despite the fact that these representations are much higher dimensional than the $IT_{sim}$ or $V1_{sim}$ representations, we nonetheless found that the joint readout from $V1_{sim}$ and $IT_{sim}$ was a better fit to human behavior (Fig. 3B).

## 5 Discussion

By simulating the visual responses properties of visual cortical neurons and using those simulations to synthesize stimuli which linearly traversed through simulations of V1 and IT, we sought to characterize the contributions of early and late ventral stream representations to perception. We found that human visual perception was sensitive both to stimuli which varied linearly through a V1-like representational space as well as to stimuli which varied linearly through an IT-like representational space.

Although this approach had the benefit of avoiding hand-selecting features, it is necessarily limited by the accuracy of the V1 and IT models that we constructed. Though these models do explain a great deal of variance in neural firing rates, they certainly do not explain 100% of the variance and furthermore, they only simulate a small number of neurons in each visual area. Our conclusions assumed that these model feature spaces are representative of the true representational space spanned by V1 and IT and would generalize to larger sets of neurons. This assumption has thus far held up

under a variety of experimental tests: for example, [24] recently demonstrated that images explicitly synthesized to maximize the response of a dCNN-based model V4 neuron were also able to drive the response of the corresponding biological neuron far above naturally occurring levels, and [27] used a similar approach to synthesize maximally exciting images for mouse V1 neurons.

However, it is certainly possible that the full representational space spanned by IT or some other late ventral stream region would be sufficient to support human perceptual sensitivity to the "V1-linear" images. This possibility suggests the necessity to make cortical measurements from high-level visual cortical regions to determine whether their representational geometry can support behavior in a task specifically designed to make use of V1-like features. In the absence of such cortical measurements, we used the representations in intermediate dCNN layers, which are significantly higher dimensional than our simulation of V1 or IT, but nonetheless found that the joint readout from V1 and IT was the best predictor of human behavior.

Our findings contribute to an ongoing debate about the tradeoffs between invariance and selectivity. Numerous studies have characterized the invariance of IT representations to category identity-preserving transforms, yet human observers can clearly access category-orthogonal features [1, 14]. One possible explanation is that the representation of low-level visual features is orthogonal to the representation of category, which might explain why IT can decode certain features. However, our results suggest that a better explanation is that perception can directly read visual information out of lower-level visual areas, such as V1. Exploring the anatomical and functional connectivity that might underlie such a readout mechanism would be useful for characterizing the link between sensory representation and perceptual experience.

Though we do not specifically explore the implications here, it is likely the case that feature-based attention plays a prominent role in any potential readout mechanism that would allow flexible access to both simple visual features from early visual cortical areas in addition to complex features, represented in later ventral stream regions. The importance of feature-based attention in selecting task-relevant stimuli from distinct cortical regions has been shown for the case of motion perception, where [28] demonstrated that human behavior is best explained by a flexible readout from V1 and MT, depending on whether subjects are cued to attend to the contrast or to the coherence of moving dots. It is possible that such a readout mechanism might also allow for flexible access to representations from both early and late stages of the ventral visual stream.

Finally, our results suggest a need to rethink the design of architecture of models of human visual recognition. dCNNs and other feedforward models use an architecture where the decision readout only extracts features from the last layer of the network. However, our results suggest that the primate brain might be able to flexibly extract features from earlier or later stages of processing, which might be more useful for encoding hierarchical features while maintaining both invariance as well as selectivity.

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
