# OpenReview forum: "V1- and IT-like representations are directly accessible to human visual perception"
_NeurIPS.cc/2021/Workshop/SVRHM — SVRHM 2021 Poster_

### Official Review · Reviewer_a9m5 · 2021-10-22
**V1 and IT representations are directly accessible to human visual perception**

**Rating:** 5
**Confidence:** 4

**Review:**

This article asks the question of which parts of the visual cortex are accessible to perception. This is an intriguing question, and has relevance to a number of questions in visual neuroscience, including about the nature of visual consciousness and the locus of perceptual learning.

I like the premise of this paper, it's clear and well-written. However, it has a technical flaw at the core of its central argument, and for that reason its core conclusion is not well supported. I don't recommend publication for this reason. With the right tweaks (suggested), this could be a candidate for publication in a higher impact venue or journal, so I encourage the authors to march forward.

**Positives**

- well-written
- clearly articulated rationale
- important question
- mostly appropriate methods

**Negatives**

*Weak critic*

The core argument of the paper relies on the fact that they've created, through an alignment procedure, a simulated population (a "critic") which can be used to create metamers of V1 and IT. I like the alignment procedure argument as a means of creating a V1-like or IT-like population. As critics, however, the simulated populations are small, comprising of only ~170 individual subunits, while the images are far higher dimensional. The result is that the so-called metamers of the starter images don't look like the starter images at all and have a lot of power in the null space of the critic. In contrast, in the original metamer paper (Freeman & Simoncelli 2011), at the foveal location, the generated images looked very much like the starter images, and the demos were visually convincing. However, looking at Supp Fig 1, I am not convinced.

I think that with such weak critics, the interpolated images (regardless of whether they used V1_sim or IT_sim as critic) are interpolated in a space which is a mixture of V1- and IT- like representations. Since their central argument relies on the fact that their metamers are done in V1 and IT representational space, I think the conclusion doesn't follow. There are simple ways to make stronger critics, and these should be considered either as controls or as replacements of their critics:

1. Using a dense critic. Replace, e.g. V1_sim with the entire pool2 layer.
2. Using a Gabor critic. Replace, e.g. V1_sim with a dense, multi-scale Gabor pyramid. The absolute response of a steerable pyramid basis (as in Freeman & Simoncelli 2011), with, perhaps, normalization, would work.
3. Clone the critics under spatial transformations to create a dense critic. For each of the V1_sim cells, create clone whose receptive fields are displaced spatially, or rotated and mirrored.

One could also consider creating metamers that are more robust to changes in scaling, rotation and position by using dynamic transformations during generation (Olah et al. 2017; https://distill.pub/2017/feature-visualization/).

**Potential improvements**

With some proper controls on the critic population (section above), this would be a strong conference paper. If the authors wish to turn this into a long-form paper, I think it would benefit from some enhancements:

*More discriminative experiments*

There may be ways to manipulate which population is used, for instance using an attentional prompt or using masking.The experiments may be strengthened by looking at the full structure of the distance metric rather than the metric structure across a single line.

*Per-subject analysis*

The average psychometric curve is an average of different weightings across different observers. Perhaps some observers are V1-like and other IT-like. It would be good to show the psychometric curves disaggregated across observers.

---

### Official Review · Reviewer_sFWQ · 2021-10-31
**promising direction of probing ventral visual readout, but conclusions overblown**

**Rating:** 7
**Confidence:** 4

**Review:**

This paper uses a VGG-19 model to simulate V1 and IT population responses and create images that are indistinguishable to one of the populations respectively. Metameric stimuli were created by interpolating between two image representations, while keeping the neural response in either model V1 or model IT constant. Human participants are able to distinguish between both interpolations which the authors use to argue that human behavior reads out of V1 *and* IT.

Pros:
* the question which cortical regions visual object recognition behavior is read out of is very timely and not studied well enough in literature
* the paper cleverly uses established in-silico simulations of the primate visual ventral stream to synthesize image metamers for V1 and IT neuronal populations
* the image synthesis procedure is validated for the model by showing that V1 and IT readouts remain at 50% for their respective metameric interpolations

Cons:
* There is an alternative hypothesis to explain the behavioral findings: the model-synthesized stimuli are not truly metameric to V1 and/or IT. In that case, neural representations in e.g. IT will be different for all stimuli and behavior might be read out of only IT while still differentiating across all stimuli. In other words, the very strong claim that "V1 and IT representations are directly accessible to human visual perception" completely rests on the goodness of the model to synthesize V1/IT metamers -- as frequently observed however, the predictivity of our current models tends to break down in stimulus distributions outside the training set used to map to neural recordings.
* Lack of a negative control: model-goodness would be better supported if the paper could show a case where stimuli are actually metameric to human behavior. E.g. synthesize images that are metameric to *both* V1 and IT to see whether behavior will be at 50% throughout. Following the above point however, I suspect this will fail.
* In general, I find that the paper severely overclaims its findings, starting with the title. Many inferences are stated factually even though alternative explanations could just as well explain the findings. None of the findings are conclusive in the absence of neural studies confirming that the synthesized stimuli are truly metameric to V1/IT. I strongly encourage the authors to more fairly discuss the nuances of the work or running appropriate controls for the currently very strongly worded inferences.

Minor points:
* why use VGG-19 and not newer model variants?
* L66: by "physically different images", do you mean different in pixel space? The word physical is a bit odd here
* Between how many different starting images are metamers created?

Despite the paper's drawbacks, I think authors as well as workshop audience could benefit from a discussion of this research direction at the workshop.

---

### Official Review · Reviewer_cSjG · 2021-10-31
**Interesting application of NN's ability to predict neural responses**

**Rating:** 8
**Confidence:** 4

**Review:**

While there have been papers showing, in the style of Yamins et al 2014, that DNN layers can predict neural responses, and some modifications give rise to better predictions, this paper provides a very interesting and workshop-relevant application of DNN's ability to predict neural responses. The paper is also well written, although in the following paragraphs I provide some issues:

1) The authors acknowledge this in the discussion, but it is worth pointing out that although NNs predict neural responses better than previous models, is an R squared value of 0.37 sufficient to make this claim?

2) Line 84: Perhaps replace w with alpha or some other symbol, because the interpolation factor might be confused with model weights.

3) Lines 89-92: "We paired these 10 images into 5 pairs, and for each pair, we interpolated between the V 1sim and ITsim responses to each image and synthesized 5 samples at 3 different intervals (25%, 50%, and 75%) between the two images. See supplementary figure 1 for more example stimuli used in this experiment." The phrasing is a little confusing, it seems that the IT sim metamers and V1sim metamers were interpolated between, but that is not what supp fig 1 says.

4) How representative is Figure 2A of the actual experiment? The two choice images in Figure 2A do not appear to be of equal brightness, so the subjects might end up doing something simpler like matching probe brightness to the options and choosing the better match. Has this been taken into account? I am also not convinced if this was done by looking at supplementary fig 1 (for example the elephant and face interpolation)

5) Since there is interpolation and a 2AFC, it might be useful to see, to check the consistency, if similar results hold for an adaptation experiment. If an observer stares at a 100% metamer and then stares at the 50/50 metamer and then is asked to make the choice, will adaptation cause a bias away from the 100% metamer?

6) In Figure 3A, the error bars for V1-linear and IT-linear are very similar in color so it is hard to make out the length of each error bar. In particular, the y-value for x=50 in the IT-linear plot should be close to 0.5 and it is somewhat lower, but it is hard to figure out the error bar to see if this is problematic.

7) Does the joint readout model assign equal weights to V1_sim and IT_sim features? If not, how does it weight them?

8) Would a residual layer between conv2 and conv4 help in making the network a better model, because it would have access to features from V1 as well as IT? It would make an interesting comparison.

---

### Official Review · Reviewer_BLLg · 2021-10-31
**An interesting behavioral paradigm but overstated conclusions**

**Rating:** 7
**Confidence:** 4

**Review:**

This paper introduces an intriguing behavioral paradigm where the perceptual representations of early and late visual stages are disentangled to ask whether perception directly uses some of the representations computed by lower-level cortical regions. The paper is clearly written and addresses a topic that is of interest to the SVRHM community. Additionally, the methods used for the experimental setup are an original and creative way to try and separate the responses of two different regions on perception.

My main concern is that the methodology used to generate the stimuli cannot fully support the conclusion that "IT is insufficient to support human recognition of visual features that span through a V1-like space". As the authors allude to in the discussion, the VGG-19 model used is known to not fully capture neural responses. Perhaps a model that better predicts IT would respond such that the linear trajectories in V1 are also fairly linear in IT? Or perhaps a weighted combination of multiple layers of the VGG network best explain IT, rather than single layers as used here? Testing some additional models to see if the predictions are the same would help support the conclusions although might still have the same problem if the models do not predict 100% of the neural response (maybe one of the CORnet models would be interesting to test, which seem to have better IT predictions, although still not perfect https://www.brain-score.org/).

The failure of the VGG-19 network as a complete model of perception is exemplified by the necessity to use the TV loss as part of the stimulus generation procedure -- although this type of regularization can make images "look" more natural it places additional constraints on the generated stimuli that are not part of the tested model and thus may hide inconsistencies (this is discussed in cited paper [18]). I am curious whether the results would hold if this loss was not applied?

Without full trust in the models it is difficult to draw sweeping conclusions from this work. That said, the experimental setup is interesting enough (and the authors acknowledge this limitation) that I recommend acceptance.

---

### Decision · Program_Chairs · 2021-11-02

Accept (Poster)